# Enhancement of the performance of wireless sensor networks using the multihop multiantenna power beacon path selection method in intelligent structures

**Ahmed Hammad**[1]*, **M. A. Mohamed**[1], **Heba M. Abdel-Atty**[2]

**1** Department of Electronics and Communications Engineering, Mansoura University, Mansoura, Egypt,
**2** Department of Electrical Engineering, Port Said University, Port Said, Egypt

* ahmed_khairt@std.mans.edu.eg

**Data Availability Statement:** All relevant data are within the paper and its Supporting information files.

## Abstract

Sensor nodes are the building blocks of wireless sensor networks (WSNs), which may gather, analyze, and transmit various types of information to a certain destination. Data collection and transmission to the destination are the main responsibilities of sensor nodes at specified time intervals. However, one of the biggest issues with WSNs is the creation of energy-efficient wireless network algorithms. In this paper, a multi-hop multi-antenna power beacon path selection (MMPS) protocol is proposed. The proposed approach consists of a source, a destination, relays, power beacons generating radio frequency (RF) signals for energy harvesting, and eavesdroppers. We used physical layer security associated with energy harvesting to protect data from eavesdroppers without requiring higher layer data encryption and enhance the energy consumption of wireless networks. The signal's broadcast strength must be high enough to allow for energy harvesting while being low enough to prevent eavesdropping. The process continues until the data reaches the destination. Comparing the performance of MMPS with those of conventional methods, MMPS enhanced the wireless network outage probability (OP) up to 99.7%, life time, energy consumption, protection from eavesdroppers, and more resistant to hardware impairments which increased the immunity up to 95%.

## Introduction

Several routing strategies have been proposed to save energy and enhance network performance. In this respect, wireless sensor networks (WSNs) have dynamic architectures, making energy-aware routing between wireless sensor nodes one of their main issues. previously [1], some authors proposed a centralized genetic-based clustering (CGC) protocol using a new concept called the onion approach, where cluster heads are elected on the basis of a genetic algorithm with three criteria. Their simulation results proved that the CGC protocol significantly extends network lifetime, decreases network energy consumption, and is capable of

**Funding:** The author(s) received no specific funding for this work.

**Competing interests:** The authors have declared that no competing interests exist.

considerably enhancing packet delivery and keeping nodes operational for longer times. However, these networks lack a specific infrastructure and are formed in inaccessible areas. Thus, constant control over these networks is not practicable. Consequently, they frequently come under attack. Meanwhile, Hatamian et al. [2] proposed a fuzzy rate controller with a congestion-aware routing technique that prioritizes data packets. On the basis of the data value, they proposed a queuing model for determining packet priority. Fuzzy logic and a greedy method were used to reduce packet loss and average energy consumption in nodes. They also proposed a method for enhancing the time, energy, and packet loss limitations while increasing the system quality of service. In a distributed environment, the latency of more upstream traffic must be tested. For instance, wireless nodes are in charge of gathering physiological data from patients and keeping track of their routine medical examinations. WSNs can also be used for interior monitoring in the business sector [3]. The application of WSNs in the military and medical areas emphasizes the significance of data transmission security in these networks. In WSNs, key management is one of the techniques used to secure information transfer between network components. In a previous study [4], sensor nodes were arranged into a binary tree, and the aggregation requests were validated using a shared key, according to the authors' discussion of a dependable tree-based data aggregation approach. Only after an acknowledgment does the request become aggregated. Meanwhile, a collaborative signal processing framework gathers data from the monitoring region. Heavy calculation and assessment are required for surveillance systems, which leads to the implementation of realistically feasible solutions. In another study [5], the proposed method's encryption procedures were built in three phases with three keys that may be changed as needed to increase secrecy and security. In the proposed scheme, the network was split into several zones controlled by area managers, which were stronger nodes with extra processing and memory. The proposed approach had better scalability, flexibility, efficiency, and lower power usage. In Alimoradi et al. [6], zones were used to describe a network. The network was organized into non-overlapping hexagonal zones. There were several sensor nodes in each zone. The two forms of communication defined by the proposed approach were intra-zone and inter-zone communication. The proposed method outperformed comparable protocols in terms of attack resistance, energy usage, alive nodes, and communication overhead. Nilsaz Dezfuli and Barati [7] presented a system where the network region was split into several square grids based on a geographical foundation to increase the lifespan and coverage of WSNs. The sensor node in each grid with the highest energy was chosen as a cluster head. The residual part of the zone may then be determined thereafter. The power used for sensing is generally insignificant when compared with the power used for reporting. Thus, creative energy-efficient reporting systems for WSNs are the focus of most research efforts. In a smart environment, various sensors are used to perform or control processes using various approaches. When Internet of things (IoT) systems and smart environments are combined, smart objects perform much better. For instance, Ghorbani Dehkordi and Barati [8] used two-phase clustering and routing as a solution for the multiobjective problem, but the division of the network environment affected the performance. Intelligent sensors deployed in target places and jointly function as a system called WSNs inspect physically dynamic quantities that are also time-sensitive and delay-tolerant or real-time and non-real-time operations. Because of the power-restricted nature of these nodes, the conventional routing protocol does not take data heterogeneity into account while establishing routing pathways. Instead, it focuses on energy-efficient routing to extend the lifetime of the system [9]. Thus, a framework was created in [10] to operate with several sensor nodes, including border, common, and gateway nodes, which conduct sensing under various workloads. One of the critical elements of the applications of sensors is the energy and power modeling of WSNs, a technique for data gathering and hierarchical network management. Thus, creating

an independent and effective network among sensor nodes is crucial to providing long network lifetimes and regulated energy consumption of networks [11].

## Motivations and related work

Physical layer security (PLS) [12] has been considered a low-cost alternative to upper-layer encryption for securing data during transmission. Secrecy performance is measured in PLS using secrecy capacity, defined as the variation between the data link channel and eavesdropping link channel capacities. Various algorithms [13–15] aimed to optimize the data rate to optimize the effectiveness of secured communication methods for data links. Security improvement strategies in underlay cognitive networks (CNs) have been suggested previously [6, 16]. The transmitters in CNs must alter their broadcast energy to meet interference limits imposed by main users. Cao et al. [17] examined the security and reliability of CNs by determining the intercept probability (IP) during the existence of an external observer and the outage probability (OP) at certified recipients. According to previous studies [18, 19], transmitting wireless sensors on the route path, such as source and relays, could decrease their transmission power to avoid IP at an active eavesdropper. Ayatollahitafti et al. [20] presented chaotic compressive sensing to handle energy efficiency and security challenges simultaneously. Wireless equipment with limited battery power must gather energy from the environment to keep going, which recently popularized energy harvesting (EH) from radio frequency (RF) in mobile ad-hoc networks [21], cognitive radio ad-hoc networks [22], and WSNs [23]. This is why noninfrastructure networks [24] have received a lot of attention as an effective approach for energy-constrained wireless networks. The EH from RF approach enables nodes to harvest their energies from RF signals. In this regard, several authors suggested secure communication algorithms using RF and EH methods in [25, 26]. The secrecy OP of simultaneous wireless communication and power transmission network comprising one base station, one required data receiver, multiple main subscribers, and several EH receivers in UCR platforms was explored in [26]. Meanwhile, Hieu et al. [27] generated formulations of OP for the proposed methods over the Rayleigh fading channel by considering that EH receivers might function as eavesdroppers. They presented the shortest path selection (SPS) protocol, random path selection (RPS) protocol, and best path selection (BPS) protocol as three unique path selection techniques, but their method has only one beacon. A rechargeable helpful jammer with a rechargeable supply was used to prevent eavesdropping in [28]. Other previous studies [29–31] presented various viable receiver architectures for simultaneous wireless information and power transfer. In [32], cooperative communication algorithms were used to minimize error rates, enhance system coverage, and improve network lifetime. Other authors investigated the wireless system performance of underlay multihop cognitive radio networks in [33, 34], where secondary users harvest energy from a beacon or a main transmitter. Previously, practically most published works considered wireless equipment hardware transceivers as ideal. However, frequency noises, amplifier amplitude nonlinearity, and I/Q imbalance were common problems in low-cost wireless nodes' physical transceivers that decrease the performance of wireless networks [35–37]. Power beacons (PBs), depending on the time switching model, were described in [38] for attaining great energy consumption and increasing the range of wireless transmission in large-scale wireless communication networks. PBs were incorporated or deployed separately from the base station [39]. In [40], the main network used several primary transmitters that acted as PBs to power the secondary network's source and relay. The EH CNs throughput performance and OP were examined and evaluated. The big modeling approach with some primary transmitters expanding to infinity was also addressed. Many investigators have lately focused on multi-hop cooperative relays to enhance network performance and

increase radio coverage (e.g., [41–43]). Compared with conventional method connection, the transmitter in multi-hop networks reduces energy consumption for data transfer. Consequently, it may be found in various applications in the real world, such as WSNs, cellular networks, IoT, vehicle or people roadside, and tracking services [44–46].

## Problem definition and contributions

According to the above literature, effective energy use with secure routing is a significant area of study interest. However, most current solutions cannot modify routing performance in response to the changing environment and constrained WSN capabilities. Furthermore, contemporary work cannot choose the next hop based on the best choice, and such methods affect the performance of routing across the whole network. Additionally, routing pathways are frequently reorganized during data relay, adding redundancy in terms of both time and communication costs. The continuous transmission of routing and management messages in the setup phase is the cause of this overhead. Consequently, investigating the field of energy efficiency with a lightweight solution to extend the lifetime of the network is necessary. All of the cited previous studies provided helpful information on the performance of EH regarding PLS, allowing system designers to make more precise recommendations. However, most of them focused on dual-hop relaying systems rather than multi-hop relaying systems. Thus, we presented one novel protocol to improve the outage performance for multi-hop multi-path decode-and-forward ultra-dense networks. There are numerous pathways between a secondary source and a secondary destination in the proposed method. One of these is chosen to transmit the source information to the receiver [36]. For packet forwarding, the source and relays on the selected channel must gather energy from the environment beacon RF waves [47, 48]. These transmitting wireless sensors must alter their signal strength in eavesdropper (V) presence to fulfill an interference limitation.

We therefore propose a multi-hop multi-antenna PB path selection (MMPS) approach to increase the e2e simultaneous channel capacity. We provide accurate and asymptotic closed-form formulations of e2e OP for the proposed scheme over the Rayleigh fading channel to measure performance. Simulations using the Monte Carlo approach are then used to confirm our conclusions.

The following are the significant contributions of this paper:

- The MMPS approach, which picks the path with the maximum end-to-end channel capacity, is proposed as a path selection method using an H number of beacons to achieve the best outage performance.

- We investigate a real-world WSN application where all hardware transmitters and receivers suffer from limitations.

- The source and relay nodes use the RF method to avoid eavesdroppers from aggregating source-collected data across several hops. Furthermore, by adjusting their transmitter power, these approved transmitters can restrict the channel capacity gained on the eavesdropping lines.

- We use a non-cooperative eavesdropping scenario, where eavesdroppers work individually.

- Over a Rayleigh fading channel, we construct closed-form equations for OP. The results of Monte Carlo simulations are provided to confirm our conclusions.

The following is an overview of the paper's structure. The system model utilized in this work is described in Section IV. Section V provides an overview of three traditional protocols.

Section VI contains the proposed protocol. Section VII presents the performance evaluation and the simulation outcomes. Finally, section VIII concludes the article.

## System model

In Fig 1, the proposed protocol is expressed by a system model, where S is the source connected with the destination D in a multi-hop approach. Further, there are N possible pathways between S and D, but just one is chosen to serve the communication between them.

Let us call the count of relays $Y_k$ (denoted by $R_{k,1}, R_{k,2}, ..., R_{k,Y_k}$) on the kth path, where $k = 1, 2, . . ., N$ and $Y_k \geq 1$. Furthermore, active V eavesdroppers are denoted by $V_1, V_2, . . ., V_u$ attempting to eavesdrop on the data sent by the source and wireless relay nodes. The RF approach in [49] is used at each hop on the specified path to stop eavesdroppers from using a maximum ratio combiner to integrate the incoming data. We assume that S and the relays, as well as all transmitters, are power-limited. Thus, they must collect the network beacon's RF energy (identified by $PB_1, PB_2, . . ., PB_H$) for data transfer. All the connections are also expected to be low-cost wireless sensor nodes with one antenna operating in half-duplex mode. Consequently, data are sent over orthogonal periods using time division multiple access. Suppose that the transmission of data is divided into orthogonal time slots and that the kth route is chosen. Specifically, in the $(i + 1)^{th}$ time slot, the relay $R_{k,i}$ sends the Z data from the source to the relay $R_{k,i+1}$, where $i = 0, 1, 2, . . ., Y_k$.

We should add that $R_{k,0} \equiv S$ and $R_{k,Y_k+1} \equiv D$ for all k. Because there are problems with the hardware, the received signal of the broadcast $R_{k,i} \to R_{k,i+1}$ and $R_{k,i} \to V_u$ can be respectively

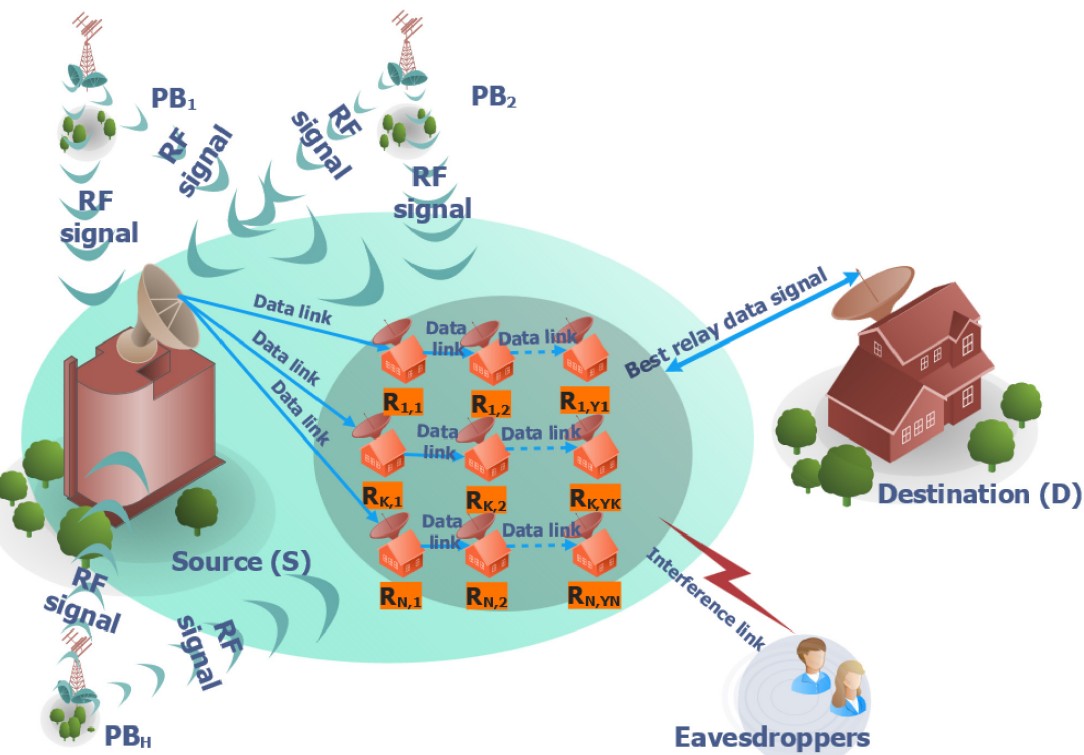

**Fig 1. System model of PB-assisted relaying protocols with relay selection methods.**

described in the following ways:

$$y_{R_{k,i}, R_{k,i+1}} = \sqrt{P_{R_{k,i}}} h_{R_{k,i} R_{k,i+1}} (Z + \eta_{R_{k,i} R_{k,i+1}}) + \mu_{R_{k,i} R_{k,i+1}} + \nu_{R_{k,i} R_{k,i+1}}, \tag{1}$$

$$y_{R_{k,i} V_u} = \sqrt{P_{R_{k,i}}} h_{R_{k,i} V_u} (Z + \eta_{R_{k,i} V_u}) + \mu_{R_{k,i} V_u} + \nu_{R_{k,i} V_u}, \tag{2}$$

where $P_{R_{k,i}}$ represents the transmitter's transmission power $R_{k,i}, h_{AB}$ is the coefficient of transmission channel from A to B link, where $A, B \in \{R_{k,i}, R_{k,i+1}, V_u\}$, $\eta_{AB}$ and $\mu_{AB}$ represent the noise made by hardware malfunctions at the transmitter A and receiver B, respectively, and $\nu_{AB}$ are Additive White Gaussian noises represented as random Gaussian variables with variance $N_0$ and zero mean.

Assuming the Rayleigh fading channel and the channel gain as $\gamma_{AB} = |h_{AB}|^2$, where $\gamma_{AB}$ is an exponential random variable, the cumulative distribution function (CDF) and probability density function (PDF) are given, respectively, as

$$F_{\gamma_{AB}}(x) = 1 - \exp((-\lambda_{AB})x), \tag{3}$$

$$f_{\gamma_{AB}}(x) = \lambda_{AB} \exp((-\lambda_{AB})x), \tag{4}$$

where $\lambda_{AB}$ is $\gamma_{AB}$ a parameter and equal to $1/\varepsilon\{\gamma_{AB}\}$ and an predicted operator is $\varepsilon\{.\}$. is formulated as in [50] to calculate the path loss:

$$\lambda_{AB} = d_{AB}^{\beta}, \tag{5}$$

where $d_{AB}$ is the distance between A and B, where the path loss scale $\beta$ is $(2 \geq \beta \leq 6)$ as in [35, 51, 52]. The distortion noises can be represented by $\eta_{AB}$ and $\mu_{AB}$, as a complex with a circular symmetry Zero-mean and variance Gaussian distribution $(\sigma_{AB}^t)^2 P_A$, and $(\sigma_{AB}^r)^2 P_A \gamma_{AB}$.

All nodes are considered to have used the same design, with the same rates of hardware limitations (i.e., $(\sigma_{AB}^t)^2 = \sigma_a^2, \sigma_{AB}^r = \sigma_b^2$.

Consequently, the instantaneous signal-to-noise ratio (SNR) of $R_{k,i} \rightarrow R_{k,i+1}$ and $R_{k,i} \rightarrow V_u$ able to implement this links is

$$SNR_{R_{k,i} R_{k,i+1}} = \frac{P_{R_{k,i}} \gamma_{R_{k,i} R_{k,i+1}}}{\kappa P_{R_{k,i}} \gamma_{R_{k,i} R_{k,i+1}} + N_0}, \tag{6}$$

$$SNR_{R_{k,i} V_u} = \frac{P_{R_{k,i}} \gamma_{R_{k,i} V_u}}{\kappa P_{R_{k,i}} \gamma_{R_{k,i} V_u} + N_0}, \tag{7}$$

where $k = \sigma_a^2 + \sigma_b^2$.

We assumed D as a block duration and the $(i + 1)th$ data transmission time slot for the kth path with time duration $\tau_k = D/(Y_k + 1)$. Time switching was used in this time slot, where the relay $R_{k,i}$ harvested its energy from beacon $PB$ at the time $\alpha\tau_k$, where $\alpha \in (0 < \alpha < 1)$ as shown in [33]. The energy harvested by $R_{k,i}$ can be formulated as

$$EH_{R_{k,i}} = \eta \alpha \tau_k P \sum_{h=1}^{H} \gamma_{R_{k,i}, PB_h}, \tag{8}$$

where $\eta$ represents the EH efficiency $0 < \eta < 1$, P is the transmission power of PB, and $\gamma_{P_B R_{k,i}}$ describes the channel gain of the $P_B \rightarrow R_{k,i}$ channel. We additionally supposed that the

connection between $P_B$ and $\gamma_{P_B R_{k,i}}$ is a Rayleigh fading channel, and the PDF and CDF of $\gamma_{P_B R_{k,i}}$ were formulated as in (3) and (4).

The data were sent using the time duration $(1 - \alpha)\tau_k$; thus, the power transmitted by the relay $R_{k,i}$ was formulated from energy conservation as

$$P_{R_{k,i}}^{\max} \leq \frac{EH_{R_{k,i}}}{(1-\alpha)\tau_k} \triangleq Z_{P_B R_{k,i}},$$

(9)

where $Z_{P_B R_{k,i}} = zP\sum_{h=1}^{H} \gamma_{R_{k,i},PB_h}$ with $z = \eta\alpha/(1-\alpha)$. The frequency ranges utilized for data transmission and EH were diverse to minimize interference. Furthermore, at each time slot, all of the nodes spent the same amount of time $\alpha\tau_k$ harvesting energy before using it to transmit data.

Assuming the transmitter $R_{k,i}$ can get the eavesdropping channel state information (CSI) of connections $R_{k,i} \rightarrow V_u$ because the eavesdroppers are active, it can decrease the quality of these links by adjusting the transmission power. Let us call $P_{R_{k,i}}^{V}$ the transmit power of $R_{k,i}$, which is modified to account for eavesdropping CSIs. The capacity of the channel between $R_{k,i}$ and $V_u$ is determined as follows:

$$C_{R_{k,i}V_u} = (1-\alpha)\tau_k \log_2\left(1 + \frac{P_{R_{k,i}}^{V}\gamma_{R_{k,i}V_u}}{kP_{R_{k,i}}^{V}\gamma_{R_{k,i}V_u} + N_0}\right),$$

(10)

Assuming that the eavesdroppers work individually, the channel capacity of the finest eavesdropper is used to calculate the eavesdropping information bit rate at the kth time slot (see [53])

$$C_{R_{k,i}V_u}^{tot} = \max_{u=1,2,...,U}\left(C_{R_{k,i}Vu}\right),$$

(11)

$$C_{R_{k,i}V_u}^{tot} = (1-\alpha)\tau_k \log_2\left(1 + \frac{P_{R_{k,i}}^{V}\varphi_{R_{k,i}max}}{kP_{R_{k,i}}^{V}\varphi_{R_{k,i}max} + N_0}\right),$$

(12)

where

$$\varphi_{R_{k,i}max} = \max_{u=1,2,...,U}\left(\gamma_{R_{k,i}Vu}\right),$$

(13)

The probability of an outage (OP) is the foundation of the maximum throughput received at the receiver side and is less than the desired rate (as indicated by $C_t h$).

We already have a set of criteria to prevent eavesdroppers from successfully decoding:

$$C_{R_{k,i}V_u}^{tot} \leqslant C_{th},$$

(14)

$$P_{R_{k,i}}^{V} \leqslant \frac{\rho_k N_0}{\varphi_{R_{k,i}max}(1 - \kappa\rho_k)}, (\kappa < 1/\rho_k)$$

(15)

where

$$\rho_k = 2^{c_{th}/(1+\alpha)\tau_k} - 1,$$

(16)

The max transmitting power of the relay $R_{k,i}$ can be formulated as (see (9) and (15))

$$P_{R_{k,i}}^{max} = \begin{cases} Z_{P_B R_{k,i}} & \text{if } \kappa \leq 1/\rho_k. \\ \\ \min\left(Z_{P_B R_{k,i}}, \dfrac{\rho_k N_0}{\varphi_{R_{k,i}max}(1 - \kappa\rho_k)}\right), & \text{otherwise.} \end{cases} \tag{17}$$

Consequently, the link $R_{k,i} \rightarrow R_{k,i+1}$ channel capacity can be formulated as

$$C_{R_{k,i}R_{k,i+1}} = \begin{cases} (1-\alpha)\tau_k \log\left(1 + \dfrac{\Delta_{1,k,i}}{\kappa\Delta_{1,k,i} + N_0}\right) & \text{if } \kappa > 1/\rho_k. \\ \\ (1-\alpha)\tau_k \log\left(1 + \dfrac{\Delta_{2,k,i}}{\kappa\Delta_{2,k,i} + N_0}\right), & \text{otherwise.} \end{cases} \tag{18}$$

where

$$\Delta_{1,k,i} = Z_{P_B R_{k,i}}\gamma_{R_{k,i}R_{k,i+1}}, \tag{19}$$

$$\Delta_{2,k,i} = min\left(Z_{P_B R_{k,i}}, \frac{\rho_k N_0}{\varphi_{R_{k,i}max}(1 - \kappa\rho_k)}\right), \tag{20}$$

Thereafter, we can formulate the e2e channel capacity of the kth path as

$$C_k^{e2e} = \min_{i=1,2,\ldots,Y_k+1}(C_{R_{k,i}R_{k,i+1}}), \tag{21}$$

## Traditional protocols

This section describes three multihop harvest-to-transmit WSNs with path selection methods. Hieu et al. [27] proposed three novel route selection techniques, namely, the SPS, RPS, and BPS protocols, to examine the effect of EH and hardware cognitive problems on the effectiveness of cooperative multihop WSNs during outages. They used a method to harvest energy similar to that in [33]. As a result, BPS is more resistant to hardware failure than RPS and SPS, and it can overcome the Rayleigh block fading on devices with poor hardware quality. The source randomly chooses one of the possible pathways in the first protocol, known as RPS, to send data to the destination. The e2e OP in this protocol may be expressed as

$$OP_{RPS} = 1/N\sum_{k=1}^{N}\Pr(C_a^{e2e} < C_{th}), \tag{22}$$

where $(a)\in\{1, 2, \ldots, N\}$

Although the RPS method is straightforward to construct, it may not give good outage performance because of the random selection. Because of the delay limitation, minimizing the number of hops on the chosen route enhances the e2e data rate. Thus, we suggest the SPS protocol as the second protocol. The SPS technique selects the route with the fewest hops number. Thus, the protocol's e2e OP is written as

$$OP_{SPS} = \Pr(C_b^{e2e} < C_{th}), \tag{23}$$

where $(b)\in\{1, 2, \ldots, N\}$

Finally, BPS selects the way that offers the most end-to-end channel capacity to maximize system performance. Mathematically, the OP of this protocol can be written as

$$\text{OP}_{\text{BPS}} = \Pr(C_c^{e2e} < C_{th}), \tag{24}$$

where $(c) \in \{1, 2, \ldots, N\}$

The e2e channel capacity of the kth path is formulated as follows:

$$C_k^{e2e} = \begin{cases} (1-\alpha)\tau_k \log_2\left(1 + \min_{i=1,2,\ldots,Y_k+1} \dfrac{\Delta_{1,k,i}}{\kappa\Delta_{1,k,i} + N_0}\right) & \text{if } \kappa > 1/\rho_k. \\[3mm] (1-\alpha)\tau_k \log_2\left(1 + \min_{i=1,2,\ldots,Y_k+1} \dfrac{\Delta_{2,k,i}}{\kappa\Delta_{2,k,i} + N_0}\right), & \text{otherwise.} \end{cases} \tag{25}$$

## Proposed MMPS protocol

We propose a new path selection method for intelligent structures based on the EH technique to extend the lifetime of the network and for suitability for WSNs or ad-hoc networks. The flowchart of the proposed protocol scheme is shown in Fig 2.

First, we evaluated the performance of MMPS, and the path with the maximum e2e channel capacity was picked. Second, we used the SNR to calculate the e2e OP and generate a through-put expression.

$$C_q^{e2e} = \max_{m=1,2,\ldots,N}(C_k^{e2e}), \tag{26}$$

where $(q) \in \{1, 2, \ldots, N\}$, and the e2e OP for MMPS can be formulated as

$$\text{OP}_{\text{MM}}^{e2e} = \Pr(C_q^{e2e} < C_{th}) = \prod_{k=1}^{N} \Pr(C_k^{e2e} < C_{th}), \tag{27}$$

$$\Pr(C_q^{e2e} < C_{th}) = \begin{cases} \Pr\left(\min_{i=1,2,\ldots,Y_k+1} \log_2\left(\dfrac{\Delta_{1,q,i}}{\kappa\Delta_{1,q,i} + N_0}\right) < \rho_q\right) & \text{if } \kappa \geqslant 1/\rho_k. \\[3mm] \Pr\left(\min_{i=1,2,\ldots,Y_k+1} \log_2\left(\dfrac{\Delta_{2,q,i}}{\kappa\Delta_{2,q,i} + N_0}\right) < \rho_q\right) & \text{otherwise} \end{cases} \tag{28}$$

where $OP_{MM}^{e2e}$ is expressed from $OP$ at the most direct route, and the exact form of $OP$ can be given as

$$OP = \begin{cases} \Pr\left(\dfrac{\Delta_{1,a,i}}{\kappa\Delta_{1,a,i} + N_0}\right) < \rho_a) & \text{if } \kappa \geqslant 1/\rho_a. \\[3mm] \Pr\left(\dfrac{\Delta_{2,a,i}}{\kappa\Delta_{2,a,i} + N_0}\right) < \rho_a) & \text{otherwise} \end{cases} \tag{29}$$

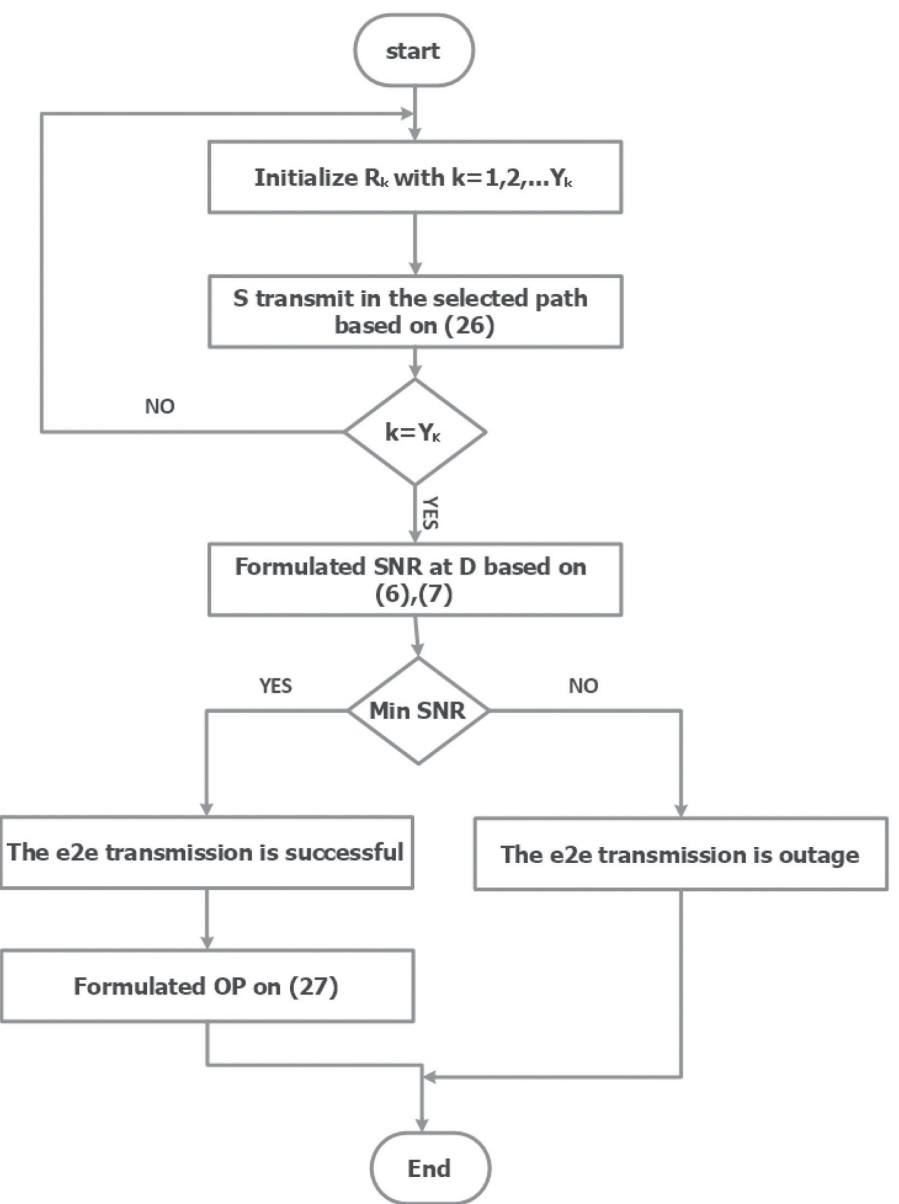

**Fig 2. The flow chart for the data transmission of (MMPS) scheme.**

## Performance evaluation

We used Monte Carlo simulations in this part to evaluate the theoretical formulas and compare our results with SPS, RPS, and BPS [27]. A matrix research facility was used to generate the simulation outcomes (MATLAB R2020a). We executed separate trials to get the e2e OP for the MMPS protocol compared with the other described protocols. In each session, we built Rayleigh channel coefficients for every link. Table 1 shows a two-dimensional grid where each coordinate is displayed. We used markers, solid lines, and dashed lines for all scenarios to represent simulation outcomes, precise theoretical results, and asymptotic theoretical findings, respectively.

**Table 1. Coordinates of network component.**

| Component | Coordinates |
|---|---|
| *Source* | $S \rightarrow (0, 0)$ |
| *Relays* | $R \rightarrow \left( \frac{i}{Y_K+1}, 0 \right)$ |
| *Destination* | $D \rightarrow (1, 0)$ |
| *Beacon* | $PB \rightarrow (x_{PB}, y_{PB})$ |
| *Eavesdroppers* | $V \rightarrow (x_V, y_V)$ |

As shown in Fig 3, we studied the effect of beacon P's transmission power (dB) on the magnitude of OP in the scenario when the eavesdroppers work individually by setting the components as shown in Table 2. As shown, there is an inverse relationship between the power of the beacon and OP, and the theoretical and simulation findings agree well. When P (dB) is low, for example, when the power in the figure is equal to −5 dB, we can detect that OP approaches 1; when the power of the beacon grows, the OP numbers decline. Thus, boosting the transmission power can help protect the physical layer versus eavesdropping assaults. In addition, when comparing the three traditional protocols, the BPS protocol clearly gets the highest OP. However, when we used the proposed MMPS protocol and added more PBs to the network by setting $H = 2$, the MMPS protocol very clearly achieves the highest OP among RPS, SPS and BPS protocols by 99.6%, 99.7% and 49.3% respectively. Furthermore, raising the transmission power of the proposed method can improve its outage performance.

Fig 4 shows the scenarios where eavesdroppers do not cooperate with the OP as a function of the number of impairments at two distinct broadcast powers of beacon P = 15 dB and the

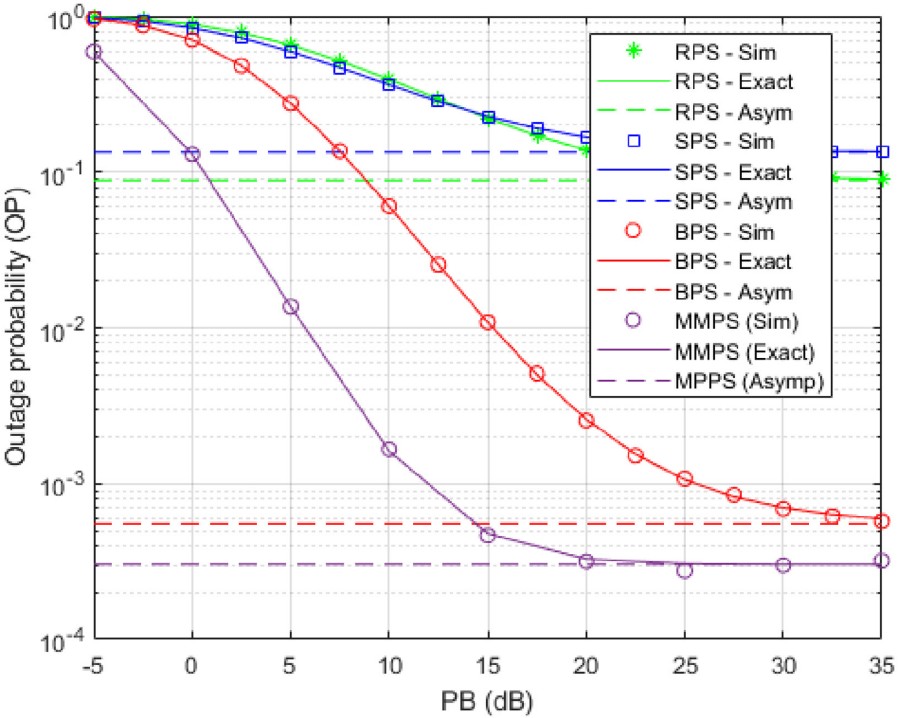

**Fig 3. OP as a function of transmitting power P in (db).**

**Table 2. Simulation conditions for each figure.**

| Component | Fig 3 | Fig 4 | Fig 5 | Fig 6 | Fig 7 |
|---|---|---|---|---|---|
| Y | [2, 3, 4] | [2, 3, 4] | [2, 3, 4] | [2, 3, 4] | [2, 3, 4] |
| R | 0.5 | 0.5 | 0.5 | 0.5 | 0.5 |
| u | 2 | 2 | 2 | 2 | 2 |
| $(x_{PB}, y_{PB})$ | (0.5,0.1) | (0.5,0.1) | (—,0.1) | (0.5,0.1) | (0.5,0.1) |
| $(x_V, y_V)$ | (0.5,1) | (0.5,1) | (0.5,1) | (0.5,—) | (0.5,1) |
| $\eta$ | 0.1 | 0.1 | 0.1 | 0.1 | 0.1 |
| $\alpha$ | 0.1 | 0.1 | 0.1 | 0.1 | - |
| H | 2 | 4 | 4 | 4 | 4 |
| $\kappa$ | 0.1 | - | 0.1 | 0.1 | 0.1 |

The mark (—) means that a certain component is variable.

wireless network is adjusted by the components shown in Table 2. The OP numbers of MMPS, BPS, RPS, and SPS rose as $\kappa$ grew, as can be seen in this graph. The MMPS still beat RPS, SPS, and BPS at all curve points. Furthermore, in high areas, the OP of all algorithms converged toward 1, specifically at $\kappa \geq 0.55$, which is consistent with the preceding section's findings. The figure also illustrates that the MMPS algorithm is much more resistant to hardware failure than RPS, SPS, and BPS by 99.5%, 99.6%, 87.7% respectively, allowing it to function efficiently on devices with poor hardware quality.

The probability of an outage is presented as a function of $x_{PB}$ in Fig 5 upon simulation by the components shown in Table 2. Apparently, when $x_{PB}$ grows to around 0.4, the outage

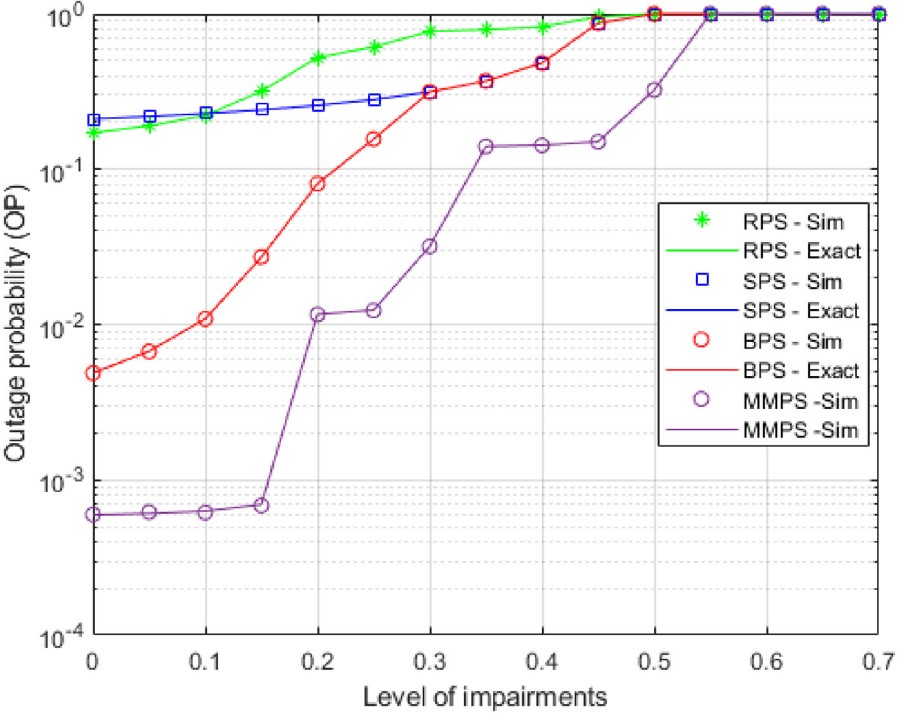

**Fig 4. OP as a function of $\kappa$.**

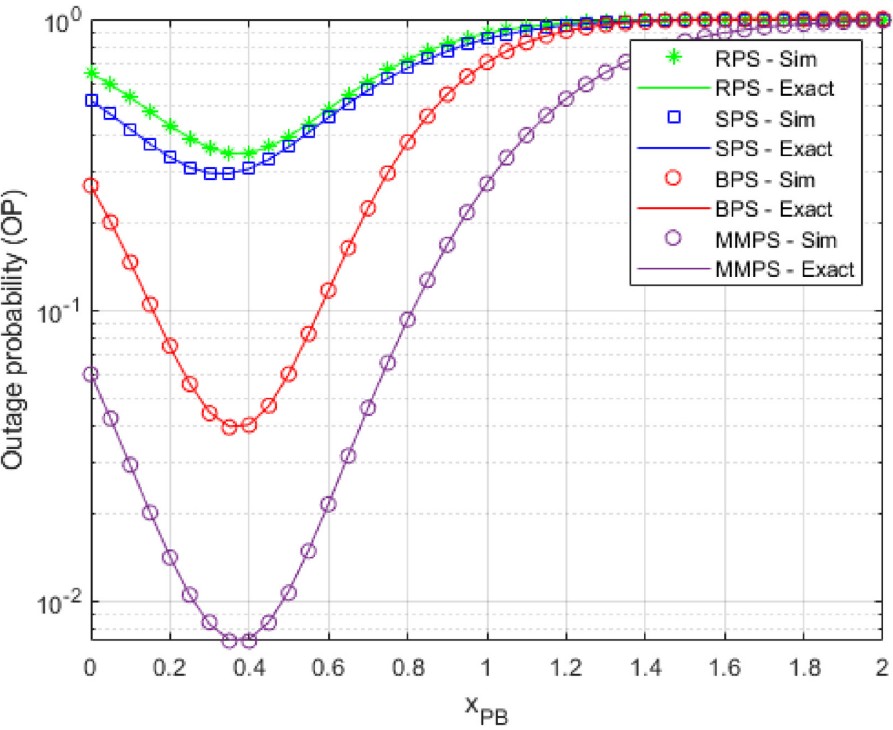

**Fig 5. OP as a function of $x_{PB}$.**

performances of the proposed algorithms improve until they reach an optimal value, beyond which they fall. Using this model, we can identify the location of the beacon where the OP achieves the ideal value. That is, once $x_{PB}$ is around 0.4, the OP of MMPS is minimized; when it is about 0.3 or 0.35, the OP values of RPS, SPS, and BPS are minimized.

In Fig 6, we investigated the impact of $y_V$ on OP upon simulation by the components shown Table 2 and set $x_{PB} = 0.4$ as the optimal value. As can be observed from the figure, OP rises when the eavesdroppers move far from the data path. Thus, when $y_V = 0$, OP has the worst effects as the connection between $V_u$ and S or $R_{k,i}$ is the smallest.

In Fig 7, using the simulation components shown in Table 2 the OP is shown as a function of the EH ratio $\alpha$ when the eavesdroppers work individually. Because it affects the received power at the ideal relay path and the transmission power of the source and relay nodes, as shown in the figure, the EH ratio $\alpha$ is significant in this situation. These graphs show the ideal value during which the OP can be decreased. Accordingly, the more energy that can be extracted from the beacon, the higher the value of $\alpha$. Consequently, relay nodes can utilize more energy to transmit data from S to D. The higher the value of $\alpha$, the less the amount of time necessary for communication $(1 - \alpha)\tau_k$ between $S \rightarrow R$ or $R \rightarrow R$. Thus, we can get the optimum outage performance when $\alpha$ reaches its optimal value. For example, the best value for MMPS equals 0.3, as shown in Fig 7.

## Conclusion

We examined the effects of hardware limitations besides EH on the throughput performance of multi-hop multi-path collaborative WSNs by presenting a unique path selection approach called the MMPS. Furthermore, we computed the proposed protocol's OP accurately and

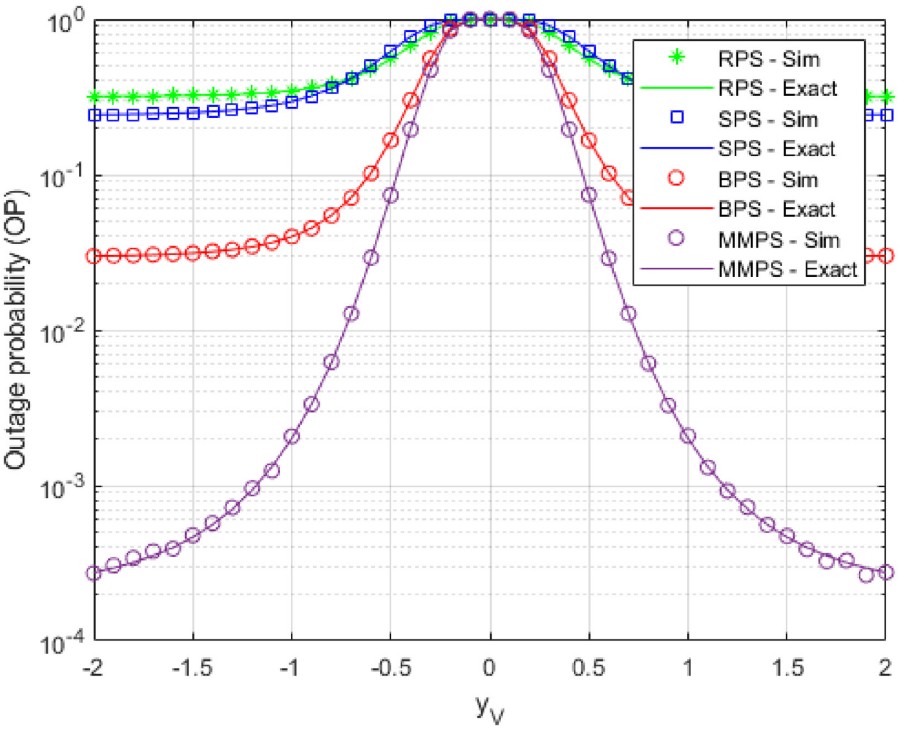

**Fig 6. OP as a function of $y_V$.**

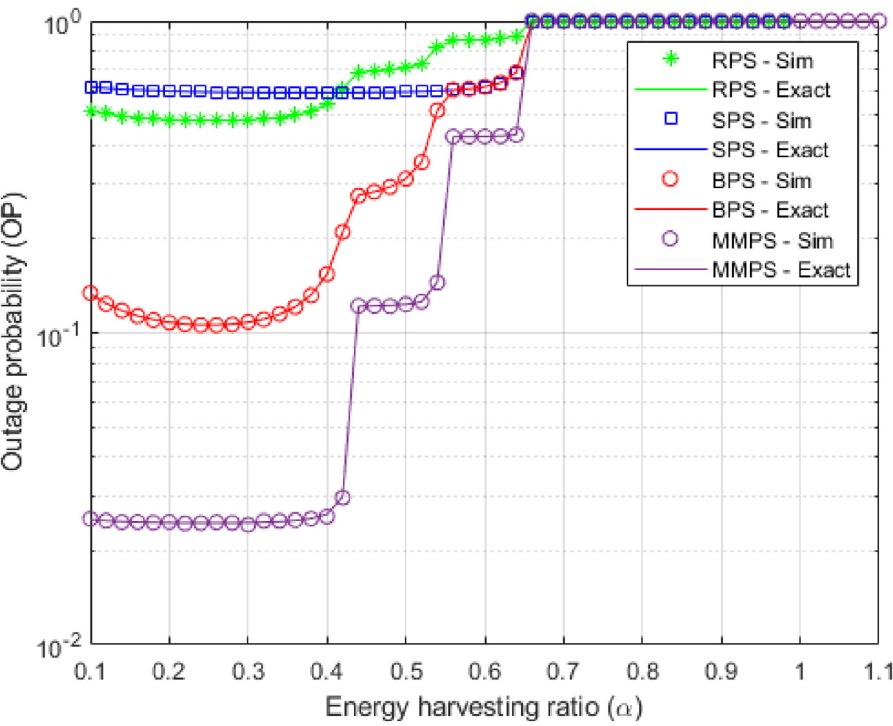

**Fig 7. OP as a function of $\alpha$.**

asymptotically in the context of several beacons, several eavesdropping assaults, and Rayleigh block fading, where the source S and wireless relay sensors can collect the beacon's RF signals. The considered EH and hardware limitation system performed securely according to simulation findings. MMPS was resistant to hardware issues, allowing it to work better with devices with poor hardware quality. Finally, the performance may be enhanced by positioning the beacon in the proper location and selecting an appropriate EH ratio $\alpha$.

## Supporting information

**S1 File.**
(DOCX)

## Author Contributions

**Conceptualization:** M. A. Mohamed, Heba M. Abdel-Atty.

**Data curation:** Ahmed Hammad, M. A. Mohamed, Heba M. Abdel-Atty.

**Formal analysis:** Ahmed Hammad, M. A. Mohamed.

**Funding acquisition:** Ahmed Hammad, Heba M. Abdel-Atty.

**Investigation:** Ahmed Hammad.

**Methodology:** M. A. Mohamed, Heba M. Abdel-Atty.

**Project administration:** M. A. Mohamed.

**Resources:** M. A. Mohamed.

**Software:** Ahmed Hammad.

**Supervision:** Ahmed Hammad, Heba M. Abdel-Atty.

**Validation:** M. A. Mohamed, Heba M. Abdel-Atty.

**Visualization:** Ahmed Hammad, M. A. Mohamed.

**Writing – original draft:** Ahmed Hammad, Heba M. Abdel-Atty.

**Writing – review & editing:** Ahmed Hammad, Heba M. Abdel-Atty.

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
