## [Decision Letter · Decision Letter 0]

31 Aug 2022

PONE-D-22-22300Enhance the Performance of Wireless Sensor Networks by Using Multi-hop Multi-antenna Power Beacons Path Selection Method in Intelligent StructuresPLOS ONE

Dear Dr. Hammad,

Thank you for submitting your manuscript to PLOS ONE. After careful consideration, we feel that it has merit but does not fully meet PLOS ONE’s publication criteria as it currently stands. Therefore, we invite you to submit a revised version of the manuscript that addresses the points raised during the review process.

We look forward to receiving your revised manuscript.

Kind regards,

Kapil Kumar Nagwanshi, PhD

Academic Editor

PLOS ONE

Reviewers' comments:

Reviewer's Responses to Questions

**Comments to the Author**

1. Is the manuscript technically sound, and do the data support the conclusions?

Reviewer #1: Yes

Reviewer #2: Yes

2. Has the statistical analysis been performed appropriately and rigorously? 

Reviewer #1: N/A

Reviewer #2: Yes

3. Have the authors made all data underlying the findings in their manuscript fully available?

Reviewer #1: Yes

Reviewer #2: Yes

4. Is the manuscript presented in an intelligible fashion and written in standard English?

Reviewer #1: Yes

Reviewer #2: Yes

5. Review Comments to the Author

Reviewer #1: This paper studied the "Enhance the Performance of Wireless Sensor Networks by Using Multi-hop Multi-antenna Power Beacons Path Selection Method in Intelligent Structures". The quality should be improved. Major revision should be done for this version of the paper as follows:

* The abstract should be rewritten. The results should be briefly stated in the abstract.

* The keyword should be written.

* The motive of the proposed method is not clear. The motivation of the proposed method should be stated in the introduction.

* Related work has not been written. In the RELATED WORK section should focus more on differences between this paper and other works to highlight the novelty of this paper. Also the disadvantages and shortcomings of the previous methods that are addressed in the proposed method must be stated.

*More achievements on this topic should be added for the Section "introductions" and "Related work" . * The cost analysis of the algorithms should be added.

* Some references are missed. At the same time, many important recent references are missing, which can support the idea of this paper, the following references should be added in the Section "References":

1- (2016). CGC: centralized genetic-based clustering protocol for wireless sensor networks using onion approach. Telecommunication systems, 62(4), 657-674.

2- (2016). Congestion-aware routing and fuzzy-based rate controller for wireless sensor networks. Radioengineering, 25(1), 114-123.

3- (2021). A reliable tree-based data aggregation method in wireless sensor networks. Peer-to-Peer Networking and Applications, 14(2), 873-887.

4- (2022). A hierarchical key management method for wireless sensor networks. Microprocessors and Microsystems, 90, 104489.

5- 2022). A hierarchical key management and authentication method for wireless sensor networks. International Journal of Communication Systems, 35(6), e5076.

6- (2019). Distributed energy efficient algorithm for ensuring coverage of wireless sensor networks. IET Communications, 13(5), 578-584.

7- (2008). A review of coverage and routing for wireless sensor networks. International Journal of Electronics and Communication Engineering, 2(1), 67-73.

8- (2022). Cluster based routing method using mobile sinks in wireless sensor network. International Journal of Electronics, 1-13.

9-(2021). EELRP: energy efficient layered routing protocol in wireless sensor networks. Computing, 103(12), 2789-2809.

10-(2021). A method for routing and data aggregating in cluster‐based wireless sensor networks. International Journal of Communication Systems, 34(7), e4754.

11- (2020). A distributed energy-efficient approach for hole repair in wireless sensor networks. Wireless Networks, 26(3), 1839-1855.

* In the Performance Evaluation section, specify what methods are compared with the proposed method.

* Mathematics modelling to analyze the method is not enough. The relative key equations should be embedded into the algorithms.

* The parameter of each equation must be described after using it. The parameters of some equations are not described.

Reviewer #2: 1. Page 1- Abstract. What is equipment malfunction? Please elaborate.

2. Page 3- Significant Contributions. Authors are required to provide a brief summary of the stated contribution: “Using not cooperate eavesdropping scenario”.

3. Page 1- Abstract. The proposed protocol improves the protection for multi-hop uncooperative wireless sensor networks. Authors need to provide an explanation for the “uncooperative wireless sensor networks”. What is the significance of multi-hop uncooperative wireless sensor networks in context of proposed research?

4. Page 8-Performance Evaluation. Authors have used Monte-Carlo simulation to evaluate the performance of proposed protocol. Authors include a few references or citations regarding the Monte Carlo simulation.

5. Authors can add a separate section of the “Related Works” to investigate the recent published work for proposing the approach.

6. How does the multi-hop, multi-path wireless sensor network protect against eavesdropping? Show with the help of security analysis.

6. PLOS authors have the option to publish the peer review history of their article (what does this mean?). If published, this will include your full peer review and any attached files.

Reviewer #1: No

Reviewer #2: No

---

## [Author Response · Author response to Decision Letter 0]

5 Oct 2022

Response letter for the manuscript editor

Dear professor Dr Solna Carreon Santos

Editorial Department

Journal of PLOS ONE

Subject: Submission of revised paper manuscript 

We authors would like to acknowledgement the reviewers for their critical looking at the article and constructive comments for possible publication. We also appreciate the editor for your great contribution and considered our manuscript entitled “Enhancement of the Performance of Wireless Sensor Networks Using the Multihop Multiantenna Power Beacon Path Selection Method in Intelligent Structures” (PONE-D-22-22300R1) for possible publication if the comments are well addressed and satisfied the editors and reviewers. We have carefully reviewed the comments and have revised the manuscript and abstract accordingly. Our responses are given in a point-by-point as comment and its response. We tried to address all the specific comments from the editor and reviewers. We hope the revised version is now suitable for publication and look forward to hearing from you in due course. If you need further information about it, we are waiting for you.

Sincerely

Ahmed Hammad

Department of Electronics and Communications Engineering, Mansoura University, Mansoura, Egypt P.O. Box 35516, email, ahmed_khairt@std.mans.edu.eg

On the behalf of co-authors 

Comments from Editor

We confirm that the submission contains "minimal data set”, which PLOS defines as consisting of the data set used to reach the conclusions drawn in the manuscript with related metadata and methods, and any additional data required to replicate the reported study findings in their entirety. This includes:

1) The values behind the means, standard deviations and other measures reported. These values are shown in detail in the manuscript.

2) The values used to build graphs. These values found in Table 1 and Table 2.

3) The points extracted from images for analysis. 

• Fig 3, OP as a function of transmitting power P in (db) 

• Fig 4, OP as a function of level of impairments

Comments from Reviewer 1

Comment: The abstract should be rewritten. The results should be briefly stated in the abstract.

Response: Thank you for pointing this out. We agree with this comment. Therefore, we have rewritten the abstract.

Comment: The keyword should be written.

Response: Agree. We have written the keyword to emphasize this point.

Comment: The motive of the proposed method is not clear. The motivation of the proposed method should be stated in the introduction.

Response: We agree with this and have incorporated your suggestion throughout the manuscript in section II named by Motivations and related work. 

Comment: Related work has not been written. In the RELATED WORK section should focus more on differences between this paper and other works to highlight the novelty of this paper. Also, the disadvantages and shortcomings of the previous methods that are addressed in the proposed method must be stated.

Response: We agree with this and have incorporated your suggestion throughout the manuscript in section II named by Motivations and related work.

Comment: More achievements on this topic should be added for the Section "introductions" and "Related work". 

Response: We agree with this and have incorporated your suggestion throughout the manuscript.

Comment: Some references are missed. At the same time, many important recent references are missing, which can support the idea of this paper, the following references should be added in the Section "References"

Response: We agree with this and have incorporated your suggestion throughout the manuscript.

Comment: In the Performance Evaluation section, specify what methods are compared with the proposed method.

Response: Agree. We have done to emphasize this point.

Comment: Mathematics modelling to analyze the method is not enough. The relative key equations should be embedded into the algorithms.

Response: You have raised an important point here. However, we believe that most of relative key equations are embedded into the algorithm, because the references added for the compared algorithms have all the required equations.

Comment: The parameter of each equation must be described after using it. The parameters of some equations are not described.

Response: Agree. We have done to emphasize this point.

Comments from Reviewer 2

Comment: 1. Page 1- Abstract. What is equipment malfunction? Please elaborate.

Response: Most of published publications on performance evaluation make the transceiver hardware in wireless terminals perfect assumptions. However, in reality, it is suffered from phase noises, amplifier-amplitude non-linearity, and in phase and quadrature imbalance (IQI), all of which considerably worsen the performance of wireless communication systems.

Comment: 2. Page 3- Significant Contributions. Authors are required to provide a brief summary of the stated contribution: “Using not cooperate eavesdropping scenario”.

Response: Agree. We have done to emphasize this point.

Comment: 3. Page 1- Abstract. The proposed protocol improves the protection for multi-hop uncooperative wireless sensor networks. Authors need to provide an explanation for the “uncooperative wireless sensor networks”. What is the significance of multi-hop uncooperative wireless sensor networks in context of proposed research?

Response: We use a non-cooperative eavesdropping scenario, where eavesdroppers don’t work as a team, but they work individually. 

Comment: 4. Page 8-Performance Evaluation. Authors have used Monte-Carlo simulation to evaluate the performance of proposed protocol. Authors include a few references or citations regarding the Monte Carlo simulation.

Response: Agree. We have done to emphasize this point.

Comment: 5. Authors can add a separate section of the “Related Works” to investigate the recent published work for proposing the approach.

Response: Agree. We have done to emphasize this point.

Comment: 6. How does the multi-hop, multi-path wireless sensor network protect against eavesdropping? Show with the help of security analysis. 

Response: Agree. We have done to emphasize this poin

---

## [Decision Letter · Decision Letter 1]

18 Oct 2022

Enhancement of the Performance of Wireless Sensor Networks Using the Multihop Multiantenna Power Beacon Path Selection Method in Intelligent Structures

PONE-D-22-22300R1

Dear Dr. Hammad,

We’re pleased to inform you that your manuscript has been judged scientifically suitable for publication and will be formally accepted for publication once it meets all outstanding technical requirements.

Kind regards,

Kapil Kumar Nagwanshi, PhD

Academic Editor

PLOS ONE

Additional Editor Comments (optional):

Reviewers' comments:

Reviewer's Responses to Questions

**Comments to the Author**

1. If the authors have adequately addressed your comments raised in a previous round of review and you feel that this manuscript is now acceptable for publication, you may indicate that here to bypass the “Comments to the Author” section, enter your conflict of interest statement in the “Confidential to Editor” section, and submit your "Accept" recommendation.

Reviewer #1: All comments have been addressed

Reviewer #2: All comments have been addressed

2. Is the manuscript technically sound, and do the data support the conclusions?

Reviewer #1: Yes

Reviewer #2: Yes

3. Has the statistical analysis been performed appropriately and rigorously? 

Reviewer #1: Yes

Reviewer #2: Yes

4. Have the authors made all data underlying the findings in their manuscript fully available?

Reviewer #1: No

Reviewer #2: Yes

5. Is the manuscript presented in an intelligible fashion and written in standard English?

Reviewer #1: Yes

Reviewer #2: Yes

6. Review Comments to the Author

Reviewer #1: The authors have considered the concerns. Necessary corrections have been made. The article is acceptable in this form

Reviewer #2: No further changes are required.

All comments are properly addressed by the authors. The paper is accepted without revision.

7. PLOS authors have the option to publish the peer review history of their article (what does this mean?). If published, this will include your full peer review and any attached files.

Reviewer #1: No

Reviewer #2: No

---

## [Editor Report · Acceptance letter]

20 Oct 2022

PONE-D-22-22300R1 

Enhancement of the Performance of Wireless Sensor Networks Using the Multihop Multiantenna Power Beacon Path Selection Method in Intelligent Structures 

Dear Dr. Hammad:

I'm pleased to inform you that your manuscript has been deemed suitable for publication in PLOS ONE. Congratulations! Your manuscript is now with our production department. 

Kind regards, 

on behalf of

Dr. Kapil Kumar Nagwanshi 

Academic Editor

PLOS ONE